# LLM Unlearning Without an Expert Curated Dataset

**Xiaoyuan Zhu, Muru Zhang, Ollie Liu, Robin Jia, Willie Neiswanger**
University of Southern California
{xzhu9839,muruzhan}@usc.edu,me@ollieliu.com,{robinjia,neiswang}@usc.edu

## Abstract

Modern large language models often encode sensitive, harmful, or copyrighted knowledge, raising the need for post-hoc unlearning—the ability to remove specific domains of knowledge from a model without full retraining. A major bottleneck in current unlearning pipelines is constructing effective forget sets—datasets that approximate the target domain and guide the model to forget it. In this work, we introduce a scalable, automated approach to generate high-quality forget sets using language models themselves. Our method synthesizes textbook-style data through a structured prompting pipeline, requiring only a domain name as input. Through experiments on unlearning biosecurity, cybersecurity, and Harry Potter novels, we show that our synthetic method consistently outperforms the baseline alternatives and is comparable to the expert-curated datasets. Additionally, ablation studies reveal that the multi-step generation pipeline significantly boosts data diversity, which in turn improves unlearning utility. Overall, our findings suggest that synthetic datasets offer a promising path toward practical, scalable unlearning for a wide range of emerging domains without the need for manual intervention. We release our code and dataset at https://github.com/xyzhu123/Synthetic_Textbook.

## 1 Introduction

Modern language models are trained on vast online datasets from diverse sources. While being remarkably versatile, their increasing knowledge capacity and instruction following capability raise concerns on their potential misuse. For instance, a language model with sufficient biosecurity knowledge could assist biological weapon production (Sandbrink, 2023), or reveal copyrighted or private material seen during training (Eldan & Russinovich, 2023; He et al., 2024). Filtering pre-training datasets is challenging and retraining a model is prohibitively expensive, making post-hoc language model "unlearning" a critical area of research. Unlike refusal-based safety mechanisms (Bai et al., 2022; Ouyang et al., 2022), unlearning the knowledge is more robust to adversarial jailbreaking (Zou et al., 2023; Wei et al., 2023), simply incapable of giving the desired answer to harmful queries. Given a target domain, the goal of unlearning is to remove the relevant knowledge from the model while preserving general capabilities.

Recent unlearning methods (Gandikota et al., 2024; Zou et al., 2024; Zhang et al., 2024) fine-tune the model to unlearn the target knowledge, which relies on a high-quality "forget set"—a dataset representative of the knowledge to be removed. Constructing the forget sets involves human labor to carefully search, collect, and filter for corpora related to the target domains. For example, WMDP (Li et al., 2024) constructs the forget set by first defining a threat model and deciding a set of subfields and knowledge categories to focus on. It then identifies high-quality databases and filters out the relevant corpora. While being effective, the intense human involvement limits the scalability of unlearning methods since it's not clear what data sources and considerations we should use, given a new target domain to unlearn. Tamirisa et al. (2025) employs a simpler construction pipeline that scrapes from relevant resources and filters by length or keywords. In Section 3, we evaluate its cybersecurity forget set (CTFTime) and find that it significantly underperforms, leading to a large drop in the model's general capabilities. This result gives further evidence for

unlearning methods' sensitivity to forget sets and how involved human effort is crucial in previous forget set construction.

To address the forget set bottleneck, we propose an automated pipeline that uses a large language model to automatically generate forget sets. As illustrated in Figure 1, we craft a three-stage generation pipeline for GPT-4o-mini to generate textbook-style documents as a forget set. This enables efficient generation of forget sets for any target domain, as only a keyword (*e.g.*, "biosecurity") is needed in order to obtain a forget set, requiring near-zero human effort. In Section 3, we found that our synthetically generated forget set can perform comparably with the WMDP expert-curated forget sets across two models, three unlearning methods, and three target domains. We additionally compare against filtering-based forget sets, where we ask GPT-4o-mini to filter out relevant samples from The Pile (Gao et al., 2020) and TxT360 (Tang et al., 2024) and found that they underperform our synthetic sets by a large margin.

Finally, we conduct an ablation on our synthetic data generation pipeline and find that our three-stage pipeline leads to higher diversity, quantified by Self-BLEU (Zhu et al., 2018), and higher diversity is important for achieving robust unlearning performance overall. We further show that open-weight models like Mistral-7B can also generate high-quality forget sets using our pipeline, enhancing the accessibility and reproducibility of our method. Overall, our result demonstrates that LLMs possess enough knowledge and fluency to generate effective forget sets for unlearning. With a well-designed, reusable prompting framework, we eliminate the reliance on human curation to construct forget sets, which streamlines the unlearning process for unforeseen domains as new LLM risks arise.

## 2 Constructing Forget Sets with a Language Model

### 2.1 Background on LLM Unlearning

Given a target domain, the goal of LLM unlearning is to update the model so that it forgets knowledge related to that domain while preserving its overall capabilities. The unlearning methods fine-tune the model weights $\theta$ following the general formulation in Equation 1:

$$\min_{\theta} \underbrace{\mathbb{E}_{x \in \mathcal{D}_f} \left[ \ell_{adv} \left( x; \theta \right) \right]}_{\text{forget}} + \underbrace{\mathbb{E}_{x \in \mathcal{D}_r} \left[ \ell_{reg} \left( x; \theta \right) \right]}_{\text{retain}}, \tag{1}$$

where $\mathcal{D}_f$ is the forget dataset that approximates the domain to be removed, and $\mathcal{D}_r$ is the retain dataset that approximates general, non-target knowledge to preserve. The adversarial loss $\ell_{adv}$ is used to degrade the model's performance on $\mathcal{D}_f$, while the regularization loss $\ell_{reg}$ is used to maintain the model's general performance by minimizing the model's error on the retain dataset.

While existing unlearning methods show promising results (Li et al., 2024; Zou et al., 2024; Tamirisa et al., 2025), they heavily depend on high-quality forget sets, which are labor-intensive and difficult to scale across new domains. To overcome this, we propose a scalable, automated alternative: using LLMs to synthesize domain-specific forget sets in a structured, textbook-style format. Our approach minimizes human effort while preserving the data relevance and diversity. We detail this generation pipeline in the next section.

### 2.2 Synthetic Textbook Generation Method

Inspired by the textbook-style synthetic training data in Gunasekar et al. (2023), we design a three-step prompting pipeline for the model to generate textbook-style data to express its knowledge of the target domain in a structural and comprehensive way. Our synthetic method only requires the user to specify the target domain, after which the forget set is generated automatically by the pipeline. This minimizes the need for user supervision.

Diversity in synthetic training data has been shown to correlate positively with supervised fine-tuning performance (Chen et al., 2024). To improve the diversity of the generated data

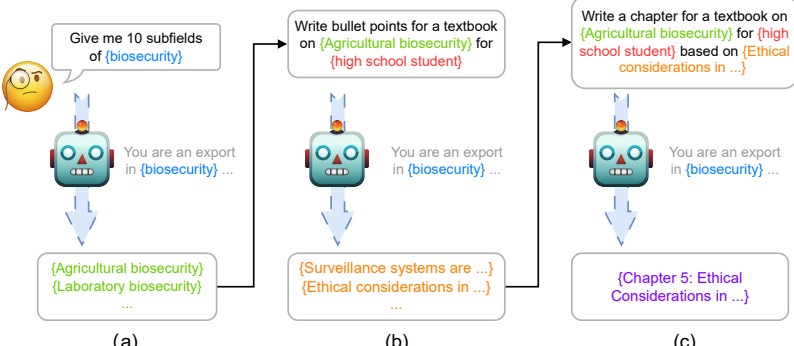

Figure 1: **Synthetic Textbook Generation Method**, consisting of three steps: (a) generating subdomains within the target domain, (b) creating bullet points tailored to the subdomain and target audience, and (c) generating textbook chapters based on the bullet points.

and provide structural guidance, we extend this approach with a three-stage generation process. As illustrated in Figure 1, the process includes: (1) generating 10 subdomains within the target domain; (2) generating 20 bullet points for each subdomain, tailored to 4 audience knowledge levels—*elementary school, high school, undergraduate, and PhD*—for a total of 800 bullet points; and (3) generating 5 textbook-style chapters per bullet point, producing 4000 chapters in total. We then split the chapters into individual sentences and select the 20,000 longest ones to form the final synthetic textbook forget set. We use GPT-4o-mini with a temperature of 0.7 throughout the pipeline. Prompts and additional details are provided in Appendix B.1. To assess the impact of each generation step, we conduct ablation by incrementally removing the three steps from the full pipeline. Evaluation results can be found in Section 4.1.

## 3 Experiments

In this section, we first present the experimental setup in Section 3.1, detailing the models and unlearning methods. Next, we describe our unlearning grid search strategy and evaluation metrics in Section 3.2. We then assess the effectiveness of the textbook synthetic method in two scenarios: unlearning hazardous knowledge (Section 3.3) and unlearning copyrighted content (Section 3.4). For hazardous knowledge, we target the biosecurity and cybersecurity domains in the WMDP benchmark (Li et al., 2024). For copyrighted knowledge, we focus on Harry Potter novels.

### 3.1 Experimental Setup

**Unlearning Method.** We adopt *Representation Misdirection for Unlearning (RMU)* (Li et al., 2024), *Representation Rerouting (RR)* (Zou et al., 2024), and *Erasure of Language Memory (ELM)* (Gandikota et al., 2024) as unlearning methods. The methods were selected because prior works (Sheshadri et al., 2024; Che et al., 2025; Fan et al., 2025) have shown that they achieve a good balance between unlearning and general performance. We also tested *Max Entropy* (Yuan et al., 2025) in preliminary experiments, but they failed to maintain model fluency after unlearning. Partial results are shown in Appendix A.2.

**Model.** We use Mistral-7B-Instruct-v0.3 (Jiang et al., 2023) and Llama3-8B-Instruct (Grattafiori et al., 2024) as target models to unlearn both WMDP and Harry Potter.

### 3.2 Grid Search and Evaluation Metrics

For each unlearning configuration, we perform a grid search over the hyperparameters of the unlearning method, which are described in Appendix A.1. We evaluate each un-

learned model by analyzing the tradeoff between removing target domain knowledge and preserving general capabilities.

We evaluate unlearning performance across three settings: the biosecurity and cybersecurity domains from the WMDP hazardous knowledge unlearning task, and the Harry Potter novels. For the first two settings, we use the biosecurity and cybersecurity subtasks from the WMDP benchmark (see Section 3.3). For the third, we use the quaternary multiple-choice Harry Potter dataset (HP MCQ) introduced by Gandikota et al. (2024) (see Section 3.4). To evaluate general capability retention in the grid search, we measure performance on tinyMMLU (Polo et al., 2024), GSM8K (Cobbe et al., 2021), and TriviaQA (Joshi et al., 2017).

To quantify the tradeoff, we define an unlearning utility $\mathcal{U}$ based on $S_f$, the percentage change in unlearning benchmark accuracy as the forgetting performance, and $S_r$, the average percentage change in general capability benchmarks. The utility is computed as:

$$\mathcal{U} = -\alpha S_f + \beta S_r, \tag{2}$$

where we use $\alpha = 0.5$ and $\beta = 0.5$ to balance the relative importance of forgetting and retention of general performance.

For each unlearning setting, we choose the top 3 hyperparameter configurations based on the unlearning utility, run them on full MMLU (Hendrycks et al., 2021), and report the average performance. We report the final unlearning utility by replacing tinyMMLU with full MMLU in the general performance retention. For the WMDP unlearning task, we also report the model accuracy on MMLU subtasks relevant to the target domain. For biosecurity, we report the College Biology and College Medicine; for cybersecurity, we report the Computer Security and College Computer Science.

In the unlearning result tables in the following sections, **General Cap.** Δ denotes $S_r$, the average percentage change in GSM8K, TriviaQA, and full MMLU scores, while **Unlearn Utility** denotes $\mathcal{U}$, the weighted tradeoff between forgetting performance ($S_f$) and retention performance ($S_r$). In each setting, the results with the best and second-best unlearn utilities are denoted in violet and light violet, respectively.

### 3.3 Unlearning WMDP

We empirically verify the effectiveness of our synthetic textbook forget sets versus two self-constructed baselines and the expert-curated forget sets from WMDP (Li et al., 2024).

**Data.** Following previous work (Li et al., 2024; Gandikota et al., 2024), we use WikiText (Merity et al., 2016) as the retain set. For each of the cybersecurity and biosecurity unlearning tasks, we construct the following forget sets for comparison. Prompts and qualitative examples are provided in Appendix B.2.

- **Expert-curated dataset**: The reference dataset curated by domain experts, provided by the WMDP benchmark. We aim for our synthetic forget set to match in performance.
- **Baseline keyword-based synthetic dataset**: A simple model-generated baseline, where samples are created by naively prompting GPT-4o-mini to list key facts given a keyword representing the target domain.
- **Baseline filtering-based dataset**: An alternative approach to automatically collect forget set. Given a target domain, we use GPT-4o-mini to filter out relevant samples from The Pile (Gao et al., 2020) and TxT360 (Tang et al., 2024).

We report the unlearning results for the biosecurity and cybersecurity tasks in Table 1 and Table 2. Overall, the synthetic textbook forget set outperforms the synthetic and filtering-based baselines and more closely approaches the performance of expert-curated datasets and even surpasses them in some cases. These results suggest that textbook-style synthetic datasets offer a promising and scalable alternative to manual data curation for unlearning, especially when expert data is unavailable.

Figure 2 shows the Pareto frontiers for two of the unlearning settings. We observe that using the unlearning utility helps identify configurations that offer the best tradeoff between forgetting and retaining general capabilities for each setting.

| Model | Method | Dataset | Unlearn Utility (↑) | General Cap. Δ (↑) | WMDP (↓) | GSM8K (↑) | TriviaQA (↑) | MMLU (↑) | | |
| --- | --- | --- | --- | --- | --- | --- | --- | --- | --- | --- |
| | | | | | | | | full | Bio | Med |
| | | | | | 0.675 | 0.502 | 0.568 | 0.596 | 0.729 | 0.578 |
| Mistral | RMU | WMDP-Bio | 24.5 | -5.2 | 0.309 | 0.486 | 0.558 | 0.569 | 0.604 | 0.443 |
| | | Textbook-Bio | 22.41 | -9.43 | 0.309 | 0.476 | 0.547 | 0.584 | 0.660 | 0.536 |
| | | Filter-Bio | 20.95 | -11.15 | 0.317 | 0.451 | 0.546 | 0.514 | 0.685 | 0.547 |
| | | Keyword-Bio | 13.69 | -4.83 | 0.457 | 0.466 | 0.535 | 0.628 | 0.743 | 0.634 |
| | RR | WMDP-Bio | 29.36 | 0.27 | 0.280 | 0.487 | 0.591 | 0.594 | 0.697 | 0.568 |
| | | Textbook-Bio | 21.52 | -9.9 | 0.318 | 0.452 | 0.569 | 0.477 | 0.567 | 0.468 |
| | | Filter-Bio | 11.05 | -22.38 | 0.375 | 0.450 | 0.541 | 0.286 | 0.250 | 0.225 |
| | | Keyword-Bio | 19.23 | -5.93 | 0.375 | 0.445 | 0.541 | 0.586 | 0.678 | 0.578 |
| | ELM | WMDP-Bio | 18.79 | -14.5 | 0.323 | 0.416 | 0.510 | 0.500 | 0.463 | 0.410 |
| | | Textbook-Bio | 13.48 | -22.71 | 0.340 | 0.327 | 0.528 | 0.439 | 0.398 | 0.382 |
| | | Filter-Bio | 14.89 | -24.81 | 0.306 | 0.363 | 0.547 | 0.339 | 0.303 | 0.404 |
| | | Keyword-Bio | 8.41 | -5.81 | 0.522 | 0.457 | 0.554 | 0.561 | 0.574 | 0.561 |
| | | | | | 0.710 | 0.753 | 0.511 | 0.638 | 0.743 | 0.630 |
| Llama3 | RMU | WMDP-Bio | 28.35 | -4.22 | 0.278 | 0.664 | 0.506 | 0.597 | 0.727 | 0.572 |
| | | Textbook-Bio | 15.54 | 0.34 | 0.492 | 0.756 | 0.513 | 0.597 | 0.727 | 0.568 |
| | | Filter-Bio | 4.92 | -10.37 | 0.567 | 0.515 | 0.512 | 0.598 | 0.725 | 0.580 |
| | | Keyword-Bio | -0.02 | 0.22 | 0.712 | 0.754 | 0.513 | 0.597 | 0.722 | 0.570 |
| | RR | WMDP-Bio | 26.34 | -6.16 | 0.292 | 0.713 | 0.524 | 0.537 | 0.576 | 0.430 |
| | | Textbook-Bio | 21.39 | -4.08 | 0.377 | 0.759 | 0.490 | 0.581 | 0.734 | 0.599 |
| | | Filter-Bio | 22.87 | -16.58 | 0.268 | 0.607 | 0.525 | 0.427 | 0.294 | 0.276 |
| | | Keyword-Bio | 2.83 | 0.35 | 0.672 | 0.754 | 0.516 | 0.637 | 0.748 | 0.624 |
| | ELM | WMDP-Bio | 7.357 | -1.25 | 0.567 | 0.723 | 0.453 | 0.609 | 0.688 | 0.601 |
| | | Textbook-Bio | 12.63 | -6.13 | 0.450 | 0.706 | 0.399 | 0.509 | 0.528 | 0.393 |
| | | Filter-Bio | 12.03 | -15.79 | 0.396 | 0.493 | 0.458 | 0.553 | 0.517 | 0.384 |
| | | Keyword-Bio | 7.959 | -6.57 | 0.513 | 0.741 | 0.372 | 0.576 | 0.618 | 0.491 |

Table 1: **Biosecurity Unlearning Results.** We use the official biosecurity forget set from WMDP (WMDP-Bio) as the expert-curated set. For full MMLU, we additionally report the College Biology (Bio) and College Medicine (Med) subtasks. See Appendix B.2 for more details on forget set construction.

While synthetic and filtering-based baselines occasionally perform well, they exhibit larger performance variances and can suffer significant drops across different settings. For instance, in Figure 2, although Keyword-Cyber yields comparable unlearning curves when unlearning Mistral-7B-Instruct-v0.3 with RMU, Keyword-Bio performs noticeably worse with ELM. In contrast, the textbook datasets maintain consistent performance across settings, highlighting its strength as a stable and general-purpose approach for constructing forget sets.

### 3.4 Unlearning Harry Potter

We include the Harry Potter unlearning task (Eldan & Russinovich, 2023; Wang et al., 2024; Gandikota et al., 2024), which aims to remove the model's knowledge about the Harry Potter novels to evaluate the textbook generation method for removing copyrighted content.

**Data.** We construct the following forget sets for the Harry Potter unlearning task:

- **Expert-curated dataset (Forget-HP)**: Direct excerpts from the Harry Potter novels.

- **Textbook-HP**: Generated using our full synthetic pipeline.

- **Textbook-HP-Simplest**: A simplified variant that directly generates textbook-style chapters without intermediate steps in Figure 1.

Results in Table 3 show that Textbook-HP consistently outperforms both the direct excerpts from the novel series and the synthetic textbook dataset with no diversity-enhancing steps. This performance gap highlights two key findings. First, the multi-step generation strategy improves the quality of the synthetic forget set. Second, the textbook dataset surpasses the excerpt-based set, demonstrating that synthetic content can sometimes be more effective for unlearning than the original copyrighted material. This result supports the broader applicability of our method to remove knowledge about copyrighted content without human supervision or direct access to their underlying data.

| Model | Method | Dataset | Unlearn Utility (↑) | General Cap. Δ (↑) | WMDP (↓) | GSM8K (↑) | TriviaQA (↑) | MMLU (↑) | | |
|---|---|---|---|---|---|---|---|---|---|---|
| | | | | | | | | full | CSec | CSci |
| | | | | | 0.415 | 0.502 | 0.642 | 0.596 | 0.660 | 0.500 |
| Mistral | RMU | CTFTime | -1.39 | -37.8 | 0.270 | 0.045 | 0.440 | 0.597 | 0.653 | 0.490 |
| | | WMDP-Cyber | 19.66 | -1.84 | 0.244 | 0.482 | 0.558 | 0.597 | 0.643 | 0.493 |
| | | Textbook-Cyber | 19.46 | -2.4 | 0.244 | 0.487 | 0.542 | 0.597 | 0.647 | 0.500 |
| | | Filter-Cyber | 10.5 | -14.92 | 0.266 | 0.290 | 0.553 | 0.596 | 0.647 | 0.497 |
| | | Keyword-Cyber | 22.13 | -0.31 | 0.230 | 0.496 | 0.568 | 0.598 | 0.650 | 0.493 |
| | RR | CTFTime | 4.2 | -27 | 0.268 | 0.219 | 0.552 | 0.466 | 0.357 | 0.357 |
| | | WMDP-Cyber | 1.98 | -33.34 | 0.260 | 0.037 | 0.566 | 0.554 | 0.260 | 0.360 |
| | | Textbook-Cyber | 20.26 | -1.86 | 0.239 | 0.477 | 0.586 | 0.574 | 0.337 | 0.460 |
| | | Filter-Cyber | 17.64 | -3.17 | 0.255 | 0.468 | 0.571 | 0.576 | 0.560 | 0.343 |
| | | Keyword-Cyber | 18.73 | -2.61 | 0.249 | 0.477 | 0.560 | 0.587 | 0.517 | 0.497 |
| | ELM | CTFTime | 3.64 | -13.57 | 0.329 | 0.315 | 0.567 | 0.577 | 0.613 | 0.477 |
| | | WMDP-Cyber | 13.87 | -8.37 | 0.265 | 0.411 | 0.565 | 0.557 | 0.430 | 0.437 |
| | | Textbook-Cyber | 14.67 | -6.82 | 0.265 | 0.428 | 0.596 | 0.533 | 0.270 | 0.403 |
| | | Filter-Cyber | 7.2 | -14.88 | 0.294 | 0.374 | 0.573 | 0.477 | 0.423 | 0.448 |
| | | Keyword-Cyber | 9.12 | -4.67 | 0.320 | 0.470 | 0.569 | 0.589 | 0.423 | 0.439 |
| | | | | | 0.468 | 0.753 | 0.592 | 0.638 | 0.770 | 0.500 |
| Llama3 | RMU | CTFTime | 4.39 | -33.58 | 0.270 | 0.018 | 0.535 | 0.588 | 0.370 | 0.383 |
| | | WMDP-Cyber | 18.1 | -12.43 | 0.240 | 0.518 | 0.518 | 0.590 | 0.250 | 0.433 |
| | | Textbook-Cyber | 17.29 | -8.59 | 0.266 | 0.715 | 0.508 | 0.510 | 0.293 | 0.317 |
| | | Filter-Cyber | -6.91 | -13.96 | 0.467 | 0.756 | 0.511 | 0.368 | 0.263 | 0.260 |
| | | Keyword-Cyber | 4.06 | -6.69 | 0.399 | 0.757 | 0.512 | 0.505 | 0.303 | 0.380 |
| | RR | CTFTime | 9.13 | -28.68 | 0.248 | 0.067 | 0.549 | 0.622 | 0.267 | 0.383 |
| | | WMDP-Cyber | 22.91 | -0.4 | 0.252 | 0.737 | 0.519 | 0.633 | 0.227 | 0.393 |
| | | Textbook-Cyber | 22.6 | 0.04 | 0.257 | 0.755 | 0.514 | 0.633 | 0.350 | 0.477 |
| | | Filter-Cyber | 15.72 | -11.05 | 0.269 | 0.685 | 0.532 | 0.458 | 0.263 | 0.267 |
| | | Keyword-Cyber | 18.15 | -0.36 | 0.296 | 0.757 | 0.508 | 0.630 | 0.300 | 0.467 |
| | ELM | CTFTime | 5.992 | 1.17 | 0.330 | 0.442 | 0.448 | 0.615 | 0.710 | 0.480 |
| | | WMDP-Cyber | 13.49 | -4.69 | 0.284 | 0.629 | 0.429 | 0.583 | 0.370 | 0.410 |
| | | Textbook-Cyber | 9.788 | -2.82 | 0.359 | 0.738 | 0.479 | 0.609 | 0.510 | 0.370 |
| | | Filter-Cyber | 0.53 | 0.09 | 0.455 | 0.738 | 0.493 | 0.635 | 0.790 | 0.540 |
| | | Keyword-Cyber | 12.53 | 1.5 | 0.352 | 0.755 | 0.506 | 0.609 | 0.570 | 0.440 |

Table 2: **Cybersecurity Unlearning Results.** We use the official cybersecurity forget sets from WMDP (WMDP-Cyber) and Tamirisa et al. (2025) (CTFTime) as the expert-curated sets. For full MMLU, we additionally report accuracy on the Computer Security (CSec) and College Computer Science (CSci) subtasks. See Appendix B.2 for more details on forget set construction.

## 4 Analysis

In this section, we evaluate the textbook synthetic method through three analyses: an ablation to assess each generation step in Section 4.1, a pairwise relevance test comparing domain alignment between textbook and comparison datasets in Section 4.2, and an unlearning experiment on the self-generated textbook datasets in Section 4.3.

### 4.1 Ablation Study on Three-Step Textbook Generation Process

We hypothesize that unlearning benefits from the text diversity of forget sets, and the multi-step textbook synthetic pipeline enhances diversity. To test this, we conduct an ablation study that progressively removes bullet points (BP), audience knowledge levels (Aud), and subdomains (Sdom) from the full pipeline as shown in Figure 1. See Appendix B.3 for detailed ablation configurations. We use Self-BLEU (Zhu et al., 2018) to quantify text diversity, where higher scores indicate lower diversity.

Table 4 presents Self-BLEU and average unlearning utility across models and unlearning methods for each ablation setting. The settings with the best unlearn utility are marked in violet. Across all target domains, Self-BLEU increases as generation steps are removed, confirming that each step contributes to greater text diversity. In the cybersecurity domain, the unlearning utility remains relatively consistent across ablation variants. For biosecurity and Harry Potter, however, the full multi-step pipeline yields the highest unlearning utility among all ablations. These findings show that the structured generation pipeline consistently improves text diversity and suggest that greater diversity may enhance unlearning.

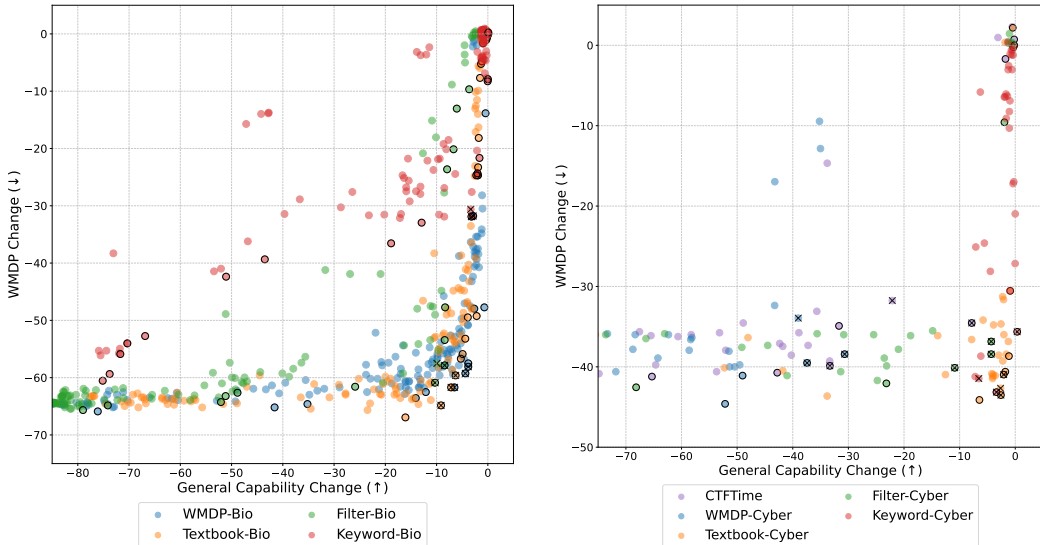

(a) Grid search results for RMU on biosecurity.     (b) Grid search results for RR on cybersecurity.

Figure 2: **Grid Search Plotting and Top-3 Point Selection.** Each panel shows the unlearning grid search for a specific method to unlearn Mistral-7B-Instruct-v0.3 on a target domain. The x-axis denotes $S_r$, the average percentage change in general capability benchmarks, and the y-axis denotes $S_f$, the percentage change in WMDP accuracy for the target domain. For each panel, the Pareto frontier points are marked with black circles, and the top 3 configurations with the highest unlearning utility are indicated with black crosses.

| Model | Method | Dataset | Unlearn Utility (↑) | General Cap. Δ (↑) | HP MCQ (↓) | GSM8K (↑) | TriviaQA (↑) | MMLU (↑) |
|---|---|---|---|---|---|---|---|---|
| | | | | | 0.717 | 0.502 | 0.568 | 0.596 |
| | RMU | Forget-HP | 13.63 | -37.9 | 0.259 | 0.478 | 0.540 | 0.596 |
| | | Textbook-HP | 31.64 | -1.28 | 0.254 | 0.489 | 0.560 | 0.597 |
| | | Textbook-HP-Simplest | 31.56 | -1.22 | 0.256 | 0.487 | 0.562 | 0.598 |
| Mistral | RR | Forget-HP | 31.17 | 0.13 | 0.271 | 0.489 | 0.586 | 0.637 |
| | | Textbook-HP | 32.62 | 0.13 | 0.271 | 0.489 | 0.586 | 0.637 |
| | | Textbook-HP-Simplest | 27.91 | -1.39 | 0.307 | 0.486 | 0.562 | 0.638 |
| | ELM | Forget-HP | 7.76 | -9.34 | 0.539 | 0.476 | 0.502 | 0.529 |
| | | Textbook-HP | 22.86 | -4.03 | 0.360 | 0.440 | 0.584 | 0.581 |
| | | Textbook-HP-Simplest | 17.21 | -5.56 | 0.430 | 0.442 | 0.585 | 0.550 |
| | | | | | 0.756 | 0.753 | 0.511 | 0.638 |
| | RMU | Forget-HP | 8.7 | -33.9 | 0.368 | 0.206 | 0.473 | 0.500 |
| | | Textbook-HP | 31.51 | -2.38 | 0.262 | 0.733 | 0.510 | 0.611 |
| | | Textbook-HP-Simplest | 21.92 | -7.85 | 0.365 | 0.744 | 0.504 | 0.504 |
| Llama3 | RR | Forget-HP | 29.37 | 0.56 | 0.316 | 0.754 | 0.520 | 0.595 |
| | | Textbook-HP | 33.41 | 0.58 | 0.255 | 0.756 | 0.511 | 0.603 |
| | | Textbook-HP-Simplest | 21.27 | -1.22 | 0.425 | 0.760 | 0.514 | 0.565 |
| | ELM | Forget-HP | 5.01 | -5.42 | 0.639 | 0.759 | 0.422 | 0.641 |
| | | Textbook-HP | 16.28 | -5.42 | 0.639 | 0.759 | 0.422 | 0.641 |
| | | Textbook-HP-Simplest | 10.17 | -6.16 | 0.556 | 0.745 | 0.423 | 0.595 |

Table 3: **Harry Potter Unlearning Results.** We use text samples from the Harry Potter novel series as the expert-curated forget set (Forget-HP). We construct two synthetic textbook variants: Textbook-HP is generated using the full synthetic method, while Textbook-HP-Simplest generates textbook-style chapters directly without any intermediate steps in Figure 1.

## 4.2   Relevance Test on Forget Sets

We hypothesize that more effective forget sets are those that are well-aligned with the target domain. To assess how well the forget sets align with the intended target domains, we perform a relevance comparison using a pairwise preference-based evaluation. Specifically, we randomly sample 5000 examples from the synthetic textbook dataset and each

| Target Domain | Ablation | Self-BLEU (↓) | Unlearn Utility (↑) |
|---|---|---|---|
| Biosecurity | Full | 0.758 | 17.83 |
| | - BP | 0.880 | 13.50 |
| | - BP & Aud | 0.899 | 12.71 |
| | - BP & Aud & Sdom | 0.930 | 14.17 |
| Cybersecurity | Full | 0.811 | 17.33 |
| | - BP | 0.880 | 17.38 |
| | - BP & Aud | 0.903 | 17.80 |
| | - BP & Aud & Sdom | 0.930 | 16.93 |
| Harry Potter | Full | 0.778 | 28.05 |
| | - BP & Aud & Sdom | 0.913 | 21.67 |

Table 4: **Self-BLEU and Average Unlearning Utility for Textbook Ablation Datasets.** See Appendix C for biosecurity and cybersecurity ablation unlearning results. Harry Potter ablation unlearning results are included in Table 3.

comparison dataset and ask two large language models—Llama3.3-70B-Instruct-Turbo and Qwen2-VL-72B-Instruct—to act as graders. Each grader is prompted to compare a pair of samples and determine which one is more relevant to the target domain. To control for length differences across datasets, we truncate all samples to the first 200 words before presenting them to the graders.

As shown in Figure 3, the textbook forget sets are preferred in most comparisons. The only exceptions are in the cybersecurity domain, where Qwen2 favors the WMDP-Cyber and Keyword-Cyber datasets. Overall, the results indicate that the synthetic textbook method generates forget data that closely matches the target domain, performing comparably or better than expert-curated alternatives.

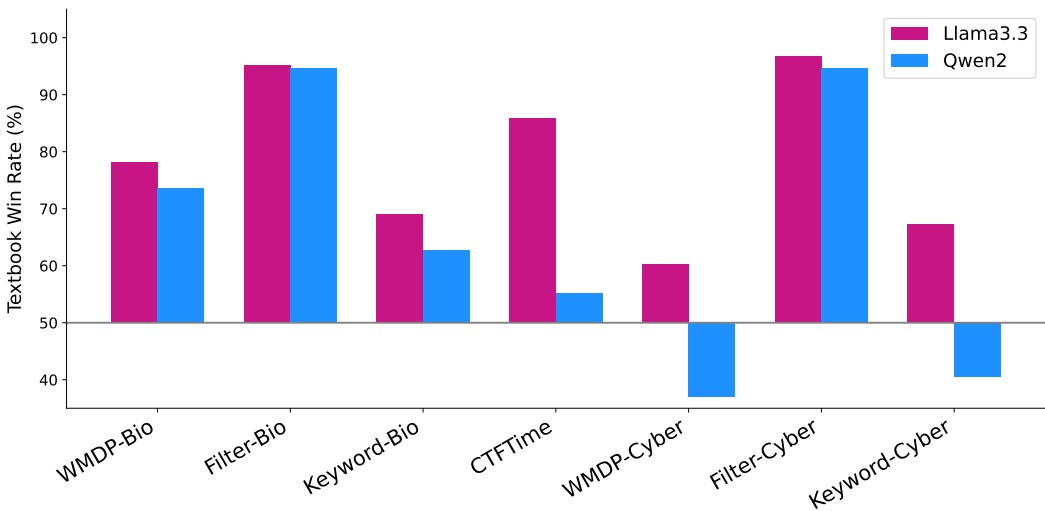

Figure 3: **Textbook Win Rates.** We perform the relevance test on the textbook dataset against the baselines in both biosecurity and cybersecurity settings. We use *Llama3.3-70B-Instruct-Turbo* and *Qwen2-VL-72B-Instruct* as graders.

## 4.3 Unlearning with Self-Generated Forget Sets

To test whether the textbook synthetic method can work without a strong external generator like GPT-4o-mini, we evaluate unlearning performance when the textbook dataset is produced by the same model that is later unlearned. For each task, we consider three variants: textbook data generated by GPT-4o-mini, by the target model itself (Mistral or Llama3), and by the other peer model. We run this experiment using RR and focus on the biosecurity and Harry Potter unlearning tasks.

As shown in Figure 4, the model-generated forget sets are effective even for smaller models. For instance, Mistral produces forget sets that perform as well as or better than that from GPT-4o-mini. While self-generated data does not always match the performance of GPT-4o-mini, as Llama3 performs worse on its own generated biosecurity set, it remains a strong alternative. The results suggest a promising trade-off, offering competitive unlearning performance along with improved reproducibility, lower cost, and ease of use.

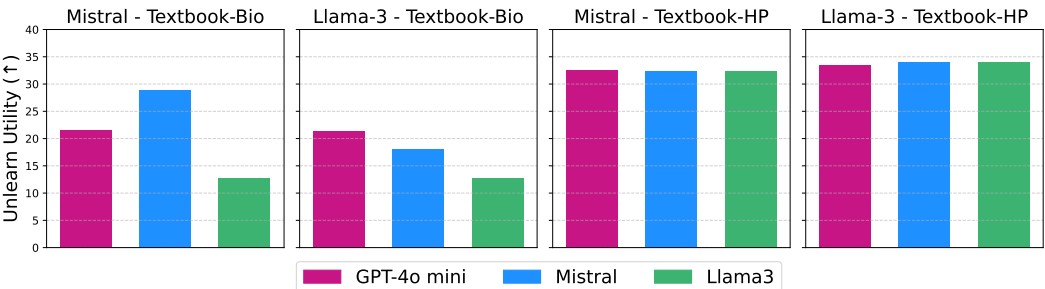

Figure 4: **Self-Generated Textbook Sets Unlearning Results.** We evaluate unlearning performance using textbook datasets generated by GPT-4o-mini, the target model itself, and the peer model. Please check the full unlearning results in Appendix C.

## 5 Related Work

**Machine unlearning** (Cao & Yang, 2015) has emerged as a powerful but lightweight paradigm to remove undesirable knowledge from trained foundation models. Most recent unlearning methods, including those featured in our experiments, perform low-rank update to remove parametric knowledge from the model (Yao et al., 2024; Zhang et al., 2024; Tamirisa et al., 2025; Li et al., 2024; Zou et al., 2024; Gandikota et al., 2024), while preserving general model performance. These methods assume access to an expert-curated *forget set*—defined as samples representative of the domain to unlearn—in order to remove concepts such as gender bias (Belrose et al., 2023), harmful behaviors (Yao et al., 2024; Liu et al., 2024), as well as fictional or hazardous knowledge (Eldan & Russinovich, 2023; Li et al., 2024). Our work aims to design generic methods that synthesize forget sets to remove arbitrary target domains, without assuming access to expert curators.

**Synthetic data generation** refers to the procedure of sourcing artificially generated data for *targeted improvements* of model behaviors (Adler et al., 2024). To date, it has driven many progresses in large language model post-training (Abdin et al., 2024a;b), especially in areas such as reasoning (Guo et al., 2025; Muennighoff et al., 2025; Lambert et al., 2024) and problem solving (Trinh et al., 2024; Chervonyi et al., 2025). In contrast, curating a *forget set* to eliminate hazardous knowledge remains an expert-in-the-loop procedure that may cost hundreds of thousands of dollars (Li et al., 2024). Our work aims to bridge this gap.

## 6 Conclusion

We present a scalable, automated framework for constructing forget sets using language models, replacing the need for expert curation with a structured, three-stage synthetic textbook generation pipeline. Our approach requires only a domain name as input and produces high-diversity, pedagogically grounded content, which we empirically demonstrate to be highly effective across multiple unlearning settings. Compared to the self-constructed baselines and expert-curated forget sets, our method achieves superior or comparable unlearning performance while preserving general model capabilities. As concerns around LLM misuse continue to grow, our results suggest a promising path toward practical, scalable unlearning for a wide range of emerging domains without the need for manual intervention.

## Acknowledgements

Thanks the USC NLP group for their helpful feedback, and Yaowen Ye and Tianyi Qiu for valuable discussions. The work was supported in part by the National Science Foundation under Grant No. IIS-2403436. Any opinions, findings, and conclusions or recommendations expressed in this material are those of the author(s) and do not necessarily reflect the views of the National Science Foundation. Additional thanks to Yiru for her personal patience, support, and encouragement.

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

# Appendix

## A    Unlearning Grid Search Details

### A.1    Hyperparameters

In all experiments, we use a fixed random seed of 42 to sample from the retain set (Wiki-Text), and multiple random seeds to sample from the forget set. Below are the detailed hyperparameters for each method:

**RMU (Li et al., 2024):**

- Layer Fine-tuning:
    - Layers: 5, 6, 7
- Alpha: 100, 1000, 10000
- Steering coefficient: 5, 50, 500
- Learning rate: `1e-5`
- Effective batch size: 4
- Steps: 50, 100
- Random seeds: 358, 23597, 2, 71
- Sample max length: 512

**ELM (Gandikota et al., 2024):**

- LoRA Fine-tuning:
    - LoRA Rank: 64
    - LoRA $\alpha$: 16
    - LoRA dropout: 0.05
- Retain loss scale: 0.1, 1, 10
- Consistency loss scale: 1
- Erase loss scale: 0.1, 1, 5
- Learning rate: `5e-5`
- Effective batch size: 8
- Steps: 800
- Random seeds: 358, 23597, 2, 71
- Sample max length: 256

**RR (Zou et al., 2024) :**

- LoRA Fine-tuning:
    - LoRA Rank: 16
    - LoRA $\alpha$: 16
    - LoRA dropout: 0.05
- LoRRA Alpha: 10
- Target layers: 10, 20
- Transform layers: all
- Learning rate: `5e-4, 1e-4, 5e-5`
- Effective batch size: 8

- Steps: 100, 200

- Random seeds: 358, 23597, 2, 71

- Sample max length: 256

## A.2   More Grid Search Results

We also provide the Max Entropy unlearning results for Mistral-7B-Instruct-v0.3 and Llama-3.1-8B-Instruct.

**Max Entropy (Yuan et al., 2025):**

- Full model fine-tuning

- Alpha: 1, 10, 100, 1000

- Learning rate: 5e-5, 1e-4, 5e-6, 1e-6

- Effective batch size: 8

- Steps: 250

- Random seeds: 358, 23597, 2, 71

- Sample max length: 2

| Model | Method | Dataset | Unlearn Utility (↑) | General Cap. Δ (↑) | WMDP (↓) | tinyMMLU (↑) | GSM8K (↑) | TriviaQA (↑) |
|-------|--------|---------|---------------------|--------------------|----------|--------------|-----------|--------------|
| | | | | | 0.675 | 0.642 | 0.502 | 0.568 |
| Mistral | Max Entropy | WMDP-Bio | 15.38 | -11.78 | 0.388 | 0.600 | 0.431 | 0.485 |
| | | Textbok-Bio | 6.05 | -15.66 | 0.488 | 0.602 | 0.423 | 0.426 |
| | | Filter-Bio | -0.29 | -0.613 | 0.675 | 0.490 | 0.642 | 0.571 |
| | | Keyword-Bio | -0.09 | -0.303 | 0.674 | 0.643 | 0.496 | 0.569 |
| | | | | | 0.728 | 0.625 | 0.772 | 0.518 |
| Llama3.1 | Max Entropy | WMDP-Bio | 7.00 | -21.35 | 0.471 | 0.591 | 0.690 | 0.270 |
| | | Textbok-Bio | 0.58 | 1.22 | 0.727 | 0.640 | 0.780 | 0.516 |
| | | Filter-Bio | 0.75 | 1.22 | 0.726 | 0.641 | 0.782 | 0.518 |
| | | Keyword-Bio | 0.62 | 0.887 | 0.726 | 0.640 | 0.781 | 0.514 |

Table 5: Biosecurity Additional Unlearning Results.

| Model | Method | Dataset | Unlearn Utility (↑) | General Cap. Δ (↑) | WMDP (↓) | tinyMMLU (↑) | GSM8K (↑) | TriviaQA (↑) |
|-------|--------|---------|---------------------|--------------------|----------|--------------|-----------|--------------|
| | | | | | 0.415 | 0.642 | 0.502 | 0.568 |
| Mistral | Max Entropy | CTFTime | 1.82 | -4.2 | 0.383 | 0.636 | 0.422 | 0.592 |
| | | WMDP-Cyber | 2.72 | -13.11 | 0.338 | 0.627 | 0.368 | 0.509 |
| | | Textbok-Cyber | 10.99 | -11.6 | 0.276 | 0.615 | 0.405 | 0.504 |
| | | Filter-Cyber | -0.11 | N/A | 0.364 | 0.614 | 0.432 | 0.456 |
| | | Keyword-Cyber | 0.11 | 0.401 | 0.628 | 0.488 | 0.541 | |
| | | | | | 0.459 | 0.625 | 0.772 | 0.518 |
| Llama3.1 | Max Entropy | CTFTime | 1.08 | 1.10 | 0.455 | 0.642 | 0.777 | 0.519 |
| | | WMDP-Cyber | 0.75 | 0.917 | 0.457 | 0.641 | 0.781 | 0.514 |
| | | Textbok-Cyber | 1.08 | 1.2 | 0.455 | 0.641 | 0.782 | 0.517 |
| | | Filter-Cyber | 0.93 | 0.653 | 0.455 | 0.641 | 0.776 | 0.518 |
| | | Keyword-Cyber | 0.75 | 0.98 | 0.457 | 0.640 | 0.781 | 0.516 |

Table 6: Cybersecurity Additional Unlearning Results.

## B  Dataset Construction Details

### B.1  Textbook Synthetic Method

| Prompt Type | Prompt Text |
|---|---|
| System | You are an expert and seasoned scholar with 20+ years of academic experience in {domain}. |
| Subdomain | Give me the names of ten subdomains of {domain}. |
| Bullet Point | You are writing an engaging, accessible, and age-appropriate textbook on {subdomain} within {domain} for {audience}. Craft a set of bullet points that cover essential concepts and practical applications. The bullet points should be clear, non-hierarchical, and self-contained. |
| Chapter | You are writing an engaging, accessible, and age-appropriate textbook on {subdomain} in {domain} for {audience}. Please write a comprehensive chapter based on the bullet point: {bullet point}. The chapter should be self-contained, on topic, and match the audience's knowledge level and understanding. |

Table 7: Prompt Templates for Synthetic Textbook Generation.

| Forget Set | Example |
|---|---|
| Textbook-Cyber | **Example 1:** Let's break down the steps involved in learning from incidents. First, after an incident occurs, the cybersecurity team will collect all the information they can about what happened. This includes looking at logs (which are like detailed diaries that compu |
| | **Example 2:** Collaboration and information sharing among organizations can further enhance the efficacy of tactical threat intelligence. By participating in threat intelligence sharing communities, organizations can pool their knowledge and experiences regarding threat |
| Textbook-Bio | **Example 1:** A more ethically sound approach to biosecurity involves adopting participatory governance models that prioritize collaboration with local communities and indigenous peoples. Such models emphasize the importance of consent and co-management, wherein local s |
| | **Example 2:** Now, let's dive into how various cultures look after their health. People around the world have unique customs and practices that help them stay well. For example, in many Asian cultures, tea is a big part of daily life. They believe that different kinds o |
| Textbook-HP | **Example 1:** The interaction between healing spells and potions in the Wizarding World provides a unique lens through which to examine the intersection of traditional and contemporary health practices. On one hand, spells represent a contemporary, almost instantaneous |
| | **Example 2:** In conclusion, the centuries-long relationship between wizards and Muggles is marked by a history of separation, prejudice, and gradually increasing understanding. While significant obstacles remain, the progress made in recent decades demonstrates that wi |

Table 8: Examples (first 256 characters) from Textbook-Bio, Textbook-Cyber, and Textbook-HP.

### B.2  Forget Set Baselines

We summarize the baseline forget sets used in our experiments across biosecurity and cybersecurity domains in Table 9. Prompt templates for Keyword-Bio and Keyword-Cyber are in Table 10; prompt templates for Filter-Cyber are in Table 11.

| Forget Set | Description |
|---|---|
| WMDP-Bio | The official biosecurity forget set in WMDP, consisting of a collection of PubMed papers related to biological threats. |
| Filter-Bio | A filtering-based subset of the WMDP biosecurity forget set proposed by Tamirisa et al. (2025), where a subset of The Pile is filtered using the keyword *"Biology"*. |
| Keyword-Bio | A synthetic dataset of 20,000 examples generated by prompting GPT-4o-mini on key facts about the keyword *biosecurity*. |
| CTFTime | The official cybersecurity forget set in Tamirisa et al. (2025), consisting of CTF writeups scraped from CTFtime. |
| WMDP-Cyber | The official cybersecurity forget set in WMDP, consisting of passages scraped via keyword search on GitHub. |
| Filter-Cyber | A filtered subset of Txt360 ArXiv, PubMed_abstract, and S2ORC_abstract datasets using the keyword *"Computer Science"*. |
| Keyword-Cyber | A synthetic dataset of 20,000 examples generated by prompting GPT-4o-mini on key facts about the keyword *cybersecurity*. |

Table 9: **Baseline forget sets used in our study.** The datasets span both real and synthetic text collections related to biosecurity and cybersecurity.

| Prompt Type | Prompt Text |
|---|---|
| System | You are an expert assistant with extensive knowledge in various domains. Provide clear, precise, and factually accurate responses based strictly on verified information relevant to the given domain. |
| Subdomain | Give me five subdomains of {domain}, separate the names with and only with "," |
| Fact | State a non-repetitive key fact in the domain of {subdomain} within {domain}, answering directly without introductory phrases or prefacing. |

Table 10: Prompt Templates for Keyword-Bio and Keyword-Cyber.

| Prompt Type | Prompt Text |
|---|---|
| System | You are a helpful assistant that classifies whether a paper falls into a specific domain. |
| Filter | Classify whether the following paper falls into the domain of {domain}. Respond with True/False.
Input: {paper_content}
Output: |

Table 11: Prompt Templates for Filter-Cyber.

## B.3 Ablation Dataset Construction

The three ablation configurations are:

- **( - BP )** removes bullet point creation. After enumerating subdomains and audience knowledge levels, we directly generate 5000 chapters by sampling 125 chapters from each of the 10 subdomains × 4 audience levels, skipping the intermediate bullet point step.

- **( - BP & Aud )** builds on (-bullet point) by also removing audience enumeration. The dataset is generated by sampling 500 chapters from each of the 10 subdomains, for a total of 5000 chapters.

- **( - BP & Aud & Sdom )** removes all intermediate steps. We directly sample 5000 chapters from the target domain without subdomain or audience specification.

## C   More Analysis Results

| Model | Method | Dataset | Unlearn Utility (↑) | General Cap. Δ (↑) | WMDP (↓) | GSM8K (↑) | TriviaQA (↑) | MMLU (↑) full | Bio | Med |
|-------|--------|---------|---------------------|--------------------|----------|-----------|--------------|---------------|-----|-----|
|       |        |         |                     |                    | 0.675    | 0.502     | 0.568        | 0.596         | 0.729 | 0.578 |
| Mistral | RMU | Textbook-Bio | 22.41 | -9.43 | 0.309 | 0.476 | 0.547 | 0.584 | 0.660 | 0.536 |
|  |  | - Bullet Point | 26.13 | -7.31 | 0.273 | 0.486 | 0.544 | 0.545 | 0.718 | 0.582 |
|  |  | - Audience | 21.27 | -7.34 | 0.338 | 0.490 | 0.545 | 0.539 | 0.699 | 0.576 |
|  |  | **- Subdomain** | 26.77 | -9.2 | 0.251 | 0.483 | 0.538 | 0.520 | 0.655 | 0.526 |
|  | RR | **Textbook-Bio** | 21.52 | -9.9 | 0.318 | 0.452 | 0.569 | 0.477 | 0.567 | 0.468 |
|  |  | - Bullet Point | 14.33 | -1.62 | 0.471 | 0.477 | 0.577 | 0.587 | 0.692 | 0.561 |
|  |  | - Audience | 17.86 | -17.82 | 0.313 | 0.361 | 0.449 | 0.458 | 0.521 | 0.439 |
|  |  | - Subdomain | 16.08 | -2.92 | 0.438 | 0.468 | 0.575 | 0.576 | 0.546 | 0.553 |
|  | ELM | **Textbook-Bio** | 13.48 | -22.7 | 0.340 | 0.327 | 0.528 | 0.439 | 0.398 | 0.382 |
|  |  | - Bullet Point | 11.37 | -14.59 | 0.423 | 0.397 | 0.563 | 0.465 | 0.472 | 0.447 |
|  |  | - Audience | 9.38 | -9.18 | 0.486 | 0.430 | 0.584 | 0.500 | 0.530 | 0.503 |
|  |  | - Subdomain | 7.27 | -6.42 | 0.533 | 0.436 | 0.587 | 0.540 | 0.618 | 0.534 |
|       |        |         |                     |                    | 0.710    | 0.753     | 0.511        | 0.638         | 0.743 | 0.630 |
| Llama-3 | RMU | Textbook-Bio | 15.54 | 0.34 | 0.492 | 0.756 | 0.513 | 0.597 | 0.727 | 0.568 |
|  |  | - Bullet Point | 13.55 | -2.16 | 0.502 | 0.758 | 0.515 | 0.549 | 0.516 | 0.574 |
|  |  | - Audience | 13.59 | -3.97 | 0.489 | 0.753 | 0.513 | 0.523 | 0.549 | 0.524 |
|  |  | **- Subdomain** | 20.29 | -1.81 | 0.409 | 0.750 | 0.517 | 0.558 | 0.493 | 0.543 |
|  | RR | **Textbook-Bio** | 21.39 | -4.08 | 0.377 | 0.759 | 0.490 | 0.581 | 0.734 | 0.599 |
|  |  | - Bullet Point | 4.02 | -0.45 | 0.650 | 0.751 | 0.510 | 0.632 | 0.748 | 0.634 |
|  |  | - Audience | 5.34 | -0.05 | 0.634 | 0.753 | 0.516 | 0.631 | 0.736 | 0.630 |
|  |  | - Subdomain | 5 | 0.24 | 0.641 | 0.755 | 0.516 | 0.635 | 0.738 | 0.628 |
|  | ELM | **Textbook-Bio** | 12.63 | -6.13 | 0.450 | 0.706 | 399 | 0.509 | 0.528 | 0.393 |
|  |  | - Bullet Point | 11.578 | -3.64 | 0.484 | 0.714 | 0.422 | 0.524 | 0.576 | 0.434 |
|  |  | - Audience | 8.847 | -1.82 | 0.554 | 0.731 | 0.470 | 0.563 | 0.660 | 0.514 |
|  |  | - Subdomain | 9.628 | 2.87 | 0.570 | 0.747 | 0.493 | 0.582 | 0.701 | 0.584 |

Table 12: Biosecurity Unlearning Results for Synthetic Method Ablation.

| Model | Method | Dataset | Unlearn Utility (↑) | General Cap. Δ (↑) | WMDP (↓) | GSM8K (↑) | TriviaQA (↑) | MMLU (↑) | | |
|---|---|---|---|---|---|---|---|---|---|---|
| | | | | | | | | full | CSec | CSci |
| | | | | | 0.415 | 0.502 | 0.568 | 0.596 | 0.660 | 0.500 |
| Mistral | RMU | Textbook-Cyber | 19.46 | -2.4 | 0.244 | 0.487 | 0.542 | 0.597 | 0.647 | 0.500 |
| | | - Bullet Point | 19.27 | -2.8 | 0.244 | 0.493 | 0.544 | 0.582 | 0.287 | 0.460 |
| | | **- Audience** | 20.44 | -2.15 | 0.237 | 0.490 | 0.556 | 0.585 | 0.307 | 0.460 |
| | | - Subdomain | 19.43 | -1.79 | 0.246 | 0.490 | 0.561 | 0.585 | 0.277 | 0.473 |
| | RR | Textbook-Cyber | 20.26 | -1.85 | 0.239 | 0.477 | 0.586 | 0.574 | 0.337 | 0.460 |
| | | - Bullet Point | 19.6 | -1.57 | 0.246 | 0.482 | 0.575 | 0.584 | 0.293 | 0.460 |
| | | - Audience | 19.33 | -1.66 | 0.248 | 0.482 | 0.576 | 0.581 | 0.300 | 0.493 |
| | | **- Subdomain** | 20.88 | -1.03 | 0.238 | 0.482 | 0.578 | 0.591 | 0.267 | 0.487 |
| | ELM | **Textbook-Cyber** | 14.67 | -6.82 | 0.265 | 0.428 | 0.596 | 0.533 | 0.270 | 0.403 |
| | | - Bullet Point | 13.92 | -6.58 | 0.272 | 0.426 | 0.588 | 0.547 | 0.305 | 0.425 |
| | | - Audience | 14.7 | -5.71 | 0.269 | 0.447 | 0.593 | 0.533 | 0.333 | 0.410 |
| | | - Subdomain | 7.47 | -5.05 | 0.332 | 0.441 | 0.583 | 0.562 | 0.467 | 0.503 |
| | | | | | 0.468 | 0.753 | 0.511 | 0.638 | 0.770 | 0.500 |
| Llama-3 | RMU | Textbook-Cyber | 17.29 | -8.59 | 0.266 | 0.715 | 0.508 | 0.510 | 0.293 | 0.317 |
| | | - Bullet Point | 18.81 | -6.45 | 0.262 | 0.749 | 0.508 | 0.522 | 0.277 | 0.290 |
| | | - Audience | 19.62 | -4.96 | 0.261 | 0.755 | 0.510 | 0.542 | 0.290 | 0.337 |
| | | **- Subdomain** | 22.35 | -2.14 | 0.249 | 0.755 | 0.514 | 0.592 | 0.300 | 0.420 |
| | RR | Textbook-Cyber | 22.60 | 0.04 | 0.257 | 0.755 | 0.514 | 0.633 | 0.350 | 0.477 |
| | | **- Bullet Point** | 22.74 | 0.100 | 0.256 | 0.755 | 0.518 | 0.630 | 0.243 | 0.487 |
| | | - Audience | 22.35 | -0.03 | 0.259 | 0.756 | 0.519 | 0.625 | 0.283 | 0.450 |
| | | - Subdomain | 20.57 | 0.03 | 0.276 | 0.754 | 0.515 | 0.632 | 0.317 | 0.483 |
| | ELM | Textbook-Cyber | 9.788 | -2.82 | 0.359 | 0.738 | 0.479 | 0.609 | 0.510 | 0.370 |
| | | - Bullet Point | 9.914 | -1.57 | 0.359 | 0.734 | 0.480 | 0.567 | 0.430 | 0.330 |
| | | - Audience | 10.38 | 0.08 | 0.366 | 0.748 | 0.499 | 0.549 | 0.380 | 9.370 |
| | | **- Subdomain** | 10.88 | -0.31 | 0.355 | 0.745 | 0.482 | 0.597 | 0.490 | 0.400 |

Table 13: Cybersecurity Unlearning Results with Synthetic Method Ablation.

| Model | Method | Dataset | Unlearn Utility (↑) | General Cap. Δ (↑) | Unlearn Metric (↓) | GSM8K (↑) | TriviaQA (↑) | MMLU (↑) | | |
|---|---|---|---|---|---|---|---|---|---|---|
| | | | | | | | | full | Bio | Med |
| | | | | | | 0.502 | 0.568 | 0.596 | 0.729 | 0.578 |
| Mistral | RR | Textbook-Bio-GPT4o | 21.52 | -9.9 | 0.318/0.675 | 0.452 | 0.569 | 0.477 | 0.567 | 0.468 |
| | | **Textbook-Bio-Mistral** | 28.95 | -1.74 | 0.272/0.675 | 0.482 | 0.577 | 0.578 | 0.667 | 0.532 |
| | | Textbook-Bio-Llama3 | 12.68 | -1.43 | 0.494/0.675 | 0.474 | 0.577 | 0.594 | 0.708 | 0.584 |
| | | **Textbook-HP-GPT4o** | 32.62 | 0.13 | 0.271/0.717 | 0.489 | 0.586 | 0.637 | / | / |
| | | Textbook-HP-Mistral | 32.36 | -0.93 | 0.247/0.717 | 0.486 | 0.565 | 0.595 | / | / |
| | | Textbook-HP-Llama3 | 32.33 | 01.18 | 0.245/0.717 | 0.487 | 0.566 | 0.567 | / | / |
| | | | | | | 0.753 | 0.511 | 0.638 | 0.743 | 0.630 |
| Llama-3 | RR | **Textbook-Bio-GPT4o** | 21.39 | -4.08 | 0.377/0.710 | 0.759 | 0.490 | 0.581 | 0.734 | 0.599 |
| | | Textbook-Bio-Mistral | 18.07 | -0.87 | 0.447/0.710 | 0.753 | 0.511 | 0.621 | 0.741 | 0.62 |
| | | Textbook-Bio-Llama3 | 12.68 | 0.1 | 0.531/0.710 | 0.759 | 0.511 | 0.635 | 0.757 | 0.636 |
| | | Textbook-HP-GPT4o | 33.41 | 0.58 | 0.255/0.756 | 0.756 | 0.511 | 0.603 | / | / |
| | | **Textbook-HP-Mistral** | 34.06 | -0.09 | 0.241/0.756 | 0.750 | 0.512 | 0.637 | / | / |
| | | Textbook-HP-Llama3 | 33.95 | 1.26 | 0.252/0.756 | 0.755 | 0.529 | 0.638 | / | / |

Table 14: **Self-Generated Forget Sets Unlearning Results**. We evaluate unlearning performance using textbook datasets generated by GPT-4o mini, the target model itself, and the peer model. All datasets are constructed using the full three-step generation process. For the biosecurity task, the unlearning metric is WMDP biology accuracy; for the Harry Potter task, it is HP MCQ accuracy.

