# OpenReview forum: "LLM Unlearning Without an Expert Curated Dataset"
_colmweb.org/COLM/2025/Conference — COLM 2025_

### Official Review · Reviewer_Smv6 · 2025-04-14

**Rating:** 6
**Confidence:** 4
**Ethics Flag:** 1

**Summary:**

This study proposes a way of creating unlearning datasets using language model themselves.
Various experiments demonstrate that synthetic datasets consistently outperform the baseline synthetic alternatives and are comparable to the expert-curated ones.
Additionally, ablation studies reveal that the multi-step generation pipeline significantly boosts data diversity, which in turn improves unlearning utility.

**Questions To Authors:**

See ``Reasons To Reject''.

**Reasons To Accept:**

- This study proposes a novel perspective for unlearning LLMs by creating synthetic datasets.
- The proposed approach is also practical.
- Experiments on wide range of benchmarks (biosecurity, cybersecurity, and Harry Potter novels) and models (Mistral, llama3) demonstrates the effectiveness of the proposed method, supporting the main claim with various experiments.
- An ablation study provides us with a new finding that higher diversity is important for achieving robust unlearning performance overall.
- The paper is well written and easy to understand.

**Reasons To Reject:**

- The proposed method is not so exciting, i.e., just creating prompting and its application (pipeline).
- It is unclear in the paper if the data samples are synthesized by target model itself, e.g., Does a Mistral model create unlearning datasets and the same model is used as target for unlearning?
- The study does not investigate if the synthesized data is reliable or not. For example, do the synthesized datasets contain hallucination? Do the synthesized datasets similar to the ground truth datasets? (e.g., semantic similarity analysis with Bert)
- A representative baseline unlearning methods: Gradient Ascent (https://arxiv.org/pdf/2210.01504) is not included in the experiment. Perhaps Negative Preference Optimization (https://arxiv.org/pdf/2404.05868) is also a major approach.

---

> ### Author Response · Authors · 2025-06-03
>
> Thank you for your insightful comments and suggestions!
>
> >  The proposed method is not so exciting, i.e., just creating prompting and its application (pipeline).
>
> Our goal is to show that synthetic datasets can be an effective and low-cost alternative to human curation. While our approach is simple, we find that it works well in practice, requires minimal user input, and is cost-efficient. For example, generating a whole textbook dataset costs around $6:
> | Domain | # of Tokens (Million) | API Cost |
> |--------|------------------------|----------|
> | Bio    |        2.44              | $6.16    |
> | Cyber  | 2.49                   | $6.27    |
> | Harry Potter     | 2.44                   | $6.15    |
>
> Our work provides a new perspective that motivates practitioners to replace human-curation effort in forget set construction with automated data generation. While a more complex synthetic data generation method can even outperform the expert-curated forget set, we leave that for future investigation.
>
> > It is unclear in the paper if the data samples are synthesized by target model itself, e.g., Does a Mistral model create unlearning datasets and the same model is used as target for unlearning?
>
> All synthetic datasets in section 3, 4.1, and 4.2 are generated by GPT-4o-mini. In Section 4.3, we instead use the target models to unlearn (Mistral-7B-Instruct-v0.3 and Llama3-8B-Instruct) to generate the forget datasets, and found that datasets from Mistral lead to competitive and sometimes better unlearning performance than GPT-4o-mini.
>
> > Do synthesized datasets contain hallucinations? Are the synthesized datasets similar to the ground truth datasets?
>
> Thank you for the question!
>
> From our human inspection, the synthesized data appears generally accurate with minimal hallucinations. However, we hope to assess its quality more directly by looking at the unlearning performance it enables. To ensure a thorough evaluation, we use a broad set of benchmarks that test domain knowledge (MMLU), math skills (GSM8K), and instruction-following (TriviaQA).
>
> >  A representative baseline unlearning method: Gradient Ascent (https://arxiv.org/pdf/2210.01504) is not included in the experiment. Perhaps Negative Preference Optimization (https://arxiv.org/pdf/2404.05868) is also a major approach.
>
> We tried Max-Entropy as a baseline because [(Yuan et al., 2024)](https://arxiv.org/pdf/2410.08109) compared it with GA and NPO, and found that Max-Entropy performed more favorably in maintaining general capabilities during unlearning. Based on that comparison, we chose to include Max-Entropy in our experiments and report the results in Appendix A.2 as it didn’t perform as well as the other three methods we evaluated in our setting. We appreciate the value of GA and NPO and see them as complementary directions.

---

> > ### Comment · Reviewer_Smv6 · 2025-06-04
> > **Response to Authors**
> >
> > Thank you very much for the response.
> > After reading the response, I have decided to maintain my score as it is.

---

### Official Review · Reviewer_zZHD · 2025-04-16

**Rating:** 7
**Confidence:** 4
**Ethics Flag:** 1

**Summary:**

This work proposes a prompting method to generate forget sets for LLM unlearning, and shows that this approach outperforms the state-of-the-art in multiple conditions.

Work description is very clear and writing is very good, reporting important concepts in a clear manner.

This work does not propose a really new and original approach: it boils down adapting data generation pipeline
to the unlearning context. However, it is well-known in the LLM community that a carefully designed engineering
contribution may have more impact than brand new original theoretical ideas;
therefore, the experimental contributions and proposed design are clearly valuable and useful for the community.

**Reasons To Accept:**

- The contribution is mostly experimental, and proposes a well-designed and successful pipeline that will prove useful for the community.
- The proposed prompting method focuses on increasing diversity of synthetic data through 3 stages, which is indeed likely the best objective, given our understanding of current literature about synthetic data for LLM.
- The experiments are nicely conducted and convincing.
- The fact to also compare with Mistral-based data generation is very welcome, as it is the only way to guarantee reproducibility, which is crucial and mandatory to qualify any study as scientific.

**Reasons To Reject:**

- The work does not propose original ideas, but a simple combination and (well-done !) adaptation of known ideas.
- A motivation is the human effort required to create the forget set: this should somehow be compared to the computational cost to create the forget set with the proposed approach, and more importantly, both minimum sizes of forget sets should be compared: indeed, the human-created forget set might be of higher quality and may be smaller than the GPT4o-created forget set to reach the same level of unlearning; and if it is the case, then the additional computational cost incurred by the proposed method does not only concern creating the forget set, but also finetuning the LLM. More generally, because the main claim is to reduce the human effort, some section about costs estimation would be welcome.
- It is clear that the proposed pipeline improves the tradeoff between unlearning and retaining general capabilities as compared to the baselines. It is no clear however, but this is a general comment that addresses many works beyond this one in the unlearning field, whether this tradeoff has become good enough to be really useful in practice.
- Larger open source LLMs could have been used to generate the forget set.

---

> ### Author Response · Authors · 2025-06-03
>
> Thank you for your helpful comments and suggestions!
>
> >  The work does not propose original ideas, but a simple combination and (well-done !) adaptation of known ideas.
>
> Our goal is to show that synthetic datasets can be an effective and low-cost alternative to human curation. While our approach is simple, we find that it works well in practice, requires minimal user input, and is cost-efficient. For example, generating a whole textbook dataset costs around $6:
>
> | Domain | # of Tokens (Million) | API Cost |
> |--------|------------------------|----------|
> | Bio    |        2.44              | $6.16    |
> | Cyber  | 2.49                   | $6.27    |
> | Harry Potter     | 2.44                   | $6.15    |
>
>
>
> > A motivation is the human effort required to create the forget set ... but also finetuning the LLM.
>
> Thank you for raising this important point.
>
> We agree it’s important to weigh the tradeoff between dataset quality and the cost of creating it. For the textbook dataset, while it only costs around $6 to generate and requires no expert intervention, our results show that the textbook datasets already perform competitively with only 200-1600 examples, and smaller  textbook datasets sometimes outperform larger expert-curated ones:
> | Model   | Method | Dataset        | Training Size   | Estimated Cost | Unlearn Utility (↑)|
> |---------|--------|----------------|-------|----------|------------------|
> | Mistral | RR     | Forget-HP       | 800       |    human effort      | 30.59         |
> | Mistral | RR     | Forget-HP       | 1600      |         human effort      | 30.30          |
> | Mistral | RR     | Textbook-HP      | 800         | $0.26     |    31.76          |
> | Mistral | RR     | Textbook-HP       | 1600      | $0.54     | 32.28          |
> | Llama3 |RR | WMDP-Cyber | 200 | human effort | 20.14 |
> | Llama3 | RR | WMDP-Cyber | 400 | human effort | 22.73 |
> | Llama3 | RR | Textbook-Cyber | 200 | $0.06| 21.91 |
> | Llama3 | RR | Textbook-Cyber | 400 | $0.12 | 21.88 |
>
>
>
>
> > It is not clear however, but this is a general comment that addresses many works beyond this one in the unlearning field, whether this tradeoff has become good enough to be really useful in practice.
>
> We agree this is an important point. Our goal of this work is to develop synthetic datasets that match the unlearning tradeoff typically achieved by expert-curated data. While it’s still an open question whether unlearning methods are strong enough yet, our results show that synthetic data can achieve competitive unlearning with comparable degradation to general capabilities such as domain knowledge, math, and instruction-following.

---

> > ### Comment · Reviewer_zZHD · 2025-06-03
> > **Read authors' answers**
> >
> > Thank you very much for your answer!
> >
> > I've read them carefully, they are indeed helpful, but do not alter enough my understanding and evaluation to make me change my opinion one way or another; so I'd like to keep my grading as it is. Thank you!

---

### Official Review · Reviewer_GsKB · 2025-05-12

**Rating:** 9
**Confidence:** 3
**Ethics Flag:** 1

**Summary:**

This study investigates how well "unlearning" (in particular, removing information about a particular topic from an LLM) can be accomplished using an automatically constructed dataset rather than a human-curated set.  The authors demonstrate the feasibility of their technique (called "Synthetic Textbook Generation") while comparing it to the human-curated forget set baseline and two other baselines: generation of sentences/facts from a keyword and extracting relevant data samples from a large corpus (in this case "The Pile") via filtering.  All the approaches use GPT-4o-mini to accomplish the generation or filtering.

Overall, the biggest takeaway is that the synthetic techniques are comparable to the unlearning using an expert-curated dataset.  By choosing three top "unlearning" methods and two strong open source LLMs, the authors steer clear of trying to provide a "best" method/model overall, and instead show that their dataset leads to stronger performance than either of the baselines and usually is very close (or exceeding) the human dataset for each technique and model.  The authors cover a breadth of topics by choosing two technical topic areas and a literary topic.  They also perform an ablation study on their (somewhat complicated) textbook generation method to show that each step adds something.  And they even ablate the GPT model by using Mistral and Llama (their two models) to do the data generation, to good effect.

There are a few downsides to this paper.  The large tables of numbers (Tables 1-3) present so much information it is hard to digest them and the interpretation of the tables (part of sections 3.3 and 3.4) is much too brief.  They do offer an "average" (Unlearn Utility) which I think is a summary of the other metrics in the table, but the explanation of it is not fully clear (where do the two S values come from?).  They also discuss Pareto frontiers but the graphs in Figure 2 are quite hard to interpret as such, and their 3 line summary is not sufficient.

I consider this paper a worthy addition to the conference and recommend it is accepted.

**Questions To Authors:**

- Is the unlearn utility (Equation 2) in fact a weighted average of the other metrics?  Can you give more details on how the S_f and S_r are calculated?
- Figure 2 - I don't see the Pareto frontier --- is it defined by the hollow markers?  Are some missing?  It is very hard to see these... maybe the graphs are just too noisy?
- what prompts are used for the filtering and keyword generation baselines?

**Reasons To Accept:**

- strong results over reasonable baselines, competitive with a human baseline
- a varied selection of methods, models, and topics
- convincing ablation study

**Reasons To Reject:**

- explanations and interpretation of results are a bit meager in some places, though the numbers in general seem to show the right trends
- reliance on GPT-4o-mini for the main technique (though this is mitigated by Section 4.3)

---

> ### Author Response · Authors · 2025-06-03
>
> Thank you for your encouraging comments and detailed suggestions!
>
> >  reliance on GPT-4o-mini for the main technique (though this is mitigated by Section 4.3)
>
> We would like to kindly highlight  section 4.3, where we show that Mistral-7B-Instruct-v0.3 produces unlearning data comparable to or better than GPT-4o-mini. This highlights that our method works with  open models. To further support reproducibility, we will open-source both the full synthetic datasets and generation scripts.
>
> > Is the unlearn utility (Equation 2) in fact a weighted average of the other metrics? Can you give more details on how the S_f and S_r are calculated?
>
> Yes, the unlearn utility is the weighted average of the other metrics. For example, in biosecurity unlearning, $S_f$ represents the percentage change in WMDP Biosecurity benchmark, and $S_r$ reflects the average percentage change on GSM8K, TriviaQA, and MMLU relative to the raw model. We will clarify the definition and calculation of it in the revised version for better readability.
>
> > Figure 2 - I don't see the Pareto frontier --- is it defined by the hollow markers? Are some missing? It is very hard to see these... maybe the graphs are just too noisy?
>
> The plot aims to display both the Pareto frontier and the top-3 runs selected by the unlearn utility. These do not always overlap. We agree the visualization in the current figure may appear noisy, and will update with  a cleaner version ([anonymized link](https://share.multcloud.link/share/7f62478c-3dd0-4952-affc-eeec218cf9df)) in the revised version of the paper. In this version, Pareto frontier points are marked with black circles, and the top-3 are indicated with black crosses.
>
> > what prompts are used for the filtering and keyword generation baselines?
>
> ### Keyword
>
> **system prompt:** *You are an expert assistant with extensive knowledge in various domains. Provide clear, precise, and factually accurate responses based strictly on verified information relevant to the given domain.*
>
> **subdomain prompt:** *Give me five subdomains of {domain}, separate the names with and only with\",\"*
>
> **fact prompt:** *State a non-repetitive key fact in the domain of {subdomain} within {domain}, answering directly without introductory phrases or prefacing.*
>
> ### Filter-Cyber
>
> **system prompt:** *You are a helpful assistant that classifies whether a paper falls into a specific domain.*
>
> **filter prompt:** *Classify whether the following paper falls into the domain of {domain}. Respond with True/False.\nInput: {paper_content}\nOutput:*

---

> > ### Comment · Reviewer_GsKB · 2025-06-09
> >
> > Thank you for the clarifications and explanations.

---

### Official Review · Reviewer_SAXR · 2025-05-13

**Rating:** 8
**Confidence:** 5
**Ethics Flag:** 1

**Summary:**

This paper proposes an automated approach to develop a target domain specific dataset that can be used in the LLM post-training phase to perform LLM unlearning. To improve safety and prevent the generation of copyrighted content, LLM unlearning step uses expert curated dataset to guide a model to forget a specific set of knowledge. Developing such forget sets is time consuming and expensive process.

The proposed method takes a step to overcome this limitation. The input to the system is the target domain name, and then the system uses LLM to generate textbook-style data using a structured prompting pipeline. Given the input domain, it produces 10 subdomains, and 20 bullet points for each subdomain tailored to 4 audience knowledge levels (elementary school, high school, undergraduate, and PhD). For each of these 800 (10x4x20) bullet points, 5 textbook-style chapters are generated. Then the chapters are split into sentences and 20K longest ones are takes to create the final forget set. GPT-4o-mini is used to generate the data.

The authors evaluated their approach by using the generated synthetic dataset in LLM unlearning, where they observed superior performance than alternative baselines and competitive performance to expert curated forget sets in WMDP. The evaluation involves two models (Mistral-7B-Instruct-v0.3, Llama3-8B-Instruct), three unlearning methods (Random Mask Unlearning, Representation Routing, and Erasure of Language Memory) and three target domains (Biosecurity and Cybersecurity from WMDP, and Harry Potter). They additionally compared against filtering-based forget sets by constructing forgets sets from The Pile and TxT360, and it underperformed the forget sets produced by the proposed approach.


* Increased my support for the paper after reading the author response.

**Reasons To Accept:**

* The proposed method is very simple but well-motivated and clearly explained.
* The data generated from the pipeline has shown high diversity, improving robustness of the unlearning process. This is a solid strength of the method.
* The experimental setup involves grid searching over parameters of the proposed method to have a set of strong baselines to compare against.
* The proposed method relies on the generation of synthetic data and the authors showed that even with smaller open source model, strong and impactful synthetic data can be generated to gain strong performance in unlearning.

**Reasons To Reject:**

* The ablation study showed that all steps are important. But, an important angle to look at is the knowledge levels' impact and how much data we really need to generate. Do we need all these four levels? Instead of generating a detailed chapter, can a sentence or expanded keyword list for the bullet points do the work? The method is really interesting, but such insights can save computational costs and help with the interpretability as well.
* A qualitative analysis to compare a handful of samples from a human-developed forget sets and the proposed synthetic data would be helpful to identify the gaps.

Minor
------
* In Table 1 caption, explain what is the 'General Cap' column, or just right 'General Capability' as the column header; and how is it computed. At a first glance it is confusing. Also add a 'lower is better' symbol to be consistant with other columns.

---

> ### Author Response · Authors · 2025-06-03
>
> Thank you for your thoughtful comments and suggestions!
>
> >  an important angle to look at is the knowledge levels' impact and how much data we really need to generate ... such insights can save computational costs and help with the interpretability as well.
>
> We agree that the cost-efficiency and interpretability of the method is important.
>
> As our ablation study shows, knowledge levels contribute to text diversity, which likely supports better unlearning. While simpler generation is an interesting direction, we found that more structured levels improve overall effectiveness.
>
> Regarding cost, the full textbook dataset generation remains relatively affordable. As shown below, the API cost per domain is around $6, which we believe is a reasonable trade-off for the quality and flexibility offered. For training cost, we standardized the token usage per training run to ensure consistent training costs across datasets.
>
> | Domain | # of Tokens (Million) | API Cost |
> |----------|------------------------|------------|
> | Bio        |        2.44                  | $6.16       |
> | Cyber    | 2.49                         | $6.27       |
> | Harry Potte | 2.44                   | $6.15       |
>
> > A qualitative analysis to compare a handful of samples from a human-developed forget sets and the proposed synthetic data would be helpful to identify the gaps.
>
> Thank you for the suggestion. We inspected the Textbook-Bio (proposed synthetic forget set)  and saw that, in comparison with WMDP-Bio (human-developed forget set), Textbook-Bio tends to use simpler language and blend educational and conversational styles.
>
> Although we are unable to share the content of WMDP-Bio due to access restrictions and usage terms (from this [link](https://huggingface.co/datasets/cais/wmdp-bio-forget-corpus) you can request access), the following are 2 examples from Textbook-Bio:
> - Furthermore, the role of genetic diversity within species should not be overlooked. Genetic diversity refers to the variation of genes within a particular species and is crucial for the adaptability of populations to changing environmental conditions. A genetically diverse population is more likely to contain individuals with traits that confer resistance to diseases or environmental stresses. Thus, preserving genetic diversity within populations is an essential aspect of biosecurity, as it enhances the overall resilience of ecosystems to invasions.
> - Effective pest management begins with accurate monitoring and identification of pest species. This step involves regular scouting of crops to observe pest populations and assess their potential impact. Identifying the specific type of pest is crucial, as different pests require different management strategies. Monitoring tools may include traps, sticky cards, and visual inspections. By establishing a baseline of pest populations, farmers can make informed decisions about when and how to intervene.
>
> We appreciate the suggestion and will include qualitative examples in the appendix to illustrate the characteristics of our generated data. We will also open-source both the generation code and full synthetic datasets to support further analysis and reproducibility.
>
> >   In Table 1 caption, explain what is the 'General Cap' column .... At a first glance it is confusing.
>
> Thanks for pointing this out. We agree that the “General Cap.” column could be clearer. This metric represents the average percentage drop in performance across general capability benchmarks (GSM8K, TriviaQA, MMLU), relative to the raw model. A good unlearning should minimize this drop while effectively removing the target knowledge. We will update the table caption and clarify the metric definition in the revised version.

---

> > ### Comment · Reviewer_SAXR · 2025-06-09
> >
> > Thank you for sharing the additional analysis. I am increasing my score.

---

### Decision · Program_Chairs · 2025-07-08

**Decision:**

Accept

**Comment:**

I have read the reviews and the discussions and I'm recommending that this paper be accepted.

The reviewers have listed the following reasons to reject the paper and I think they are all reasonable points to bring up:
- The discussion and interpretation of results is weak in some parts of the paper.
- More open source (data/weights) models could be considered.

The reviewers also list these strong reasons to accept the paper:
- The method is simple and well explained.
- Strong results over reasonable baselines.
- Compelling ablations
- The writing is clear and easy to understand.

**New point from the AC: This paper uses an unlearning method called RMU and the citation the authors give for this method is a 2024 paper by Li et al. However, that original paper by Li et al. names and describes a method called Representation Misdirection for Unlearning. In this paper, the authors say RMU stands for Random Mask Unlearning. If this is a simple mixup, it should be fixed by correcting the full name of the method or the citation.**

I think work on avoiding expensive gathering of expert data is timely and of interest to the community.